# Over-Expressing *TaSPA-B* Reduces Prolamin and Starch Accumulation in Wheat (*Triticum aestivum* L.) Grains

**DOI:** 10.3390/ijms21093257

**Published:** 2020-05-05

**Authors:** Dandan Guo, Qiling Hou, Runqi Zhang, Hongyao Lou, Yinghui Li, Yufeng Zhang, Mingshan You, Chaojie Xie, Rongqi Liang, Baoyun Li

**Affiliations:** 1Key Laboratory of Crop Heterosis and Utilization (MOE) of Ministry of Education, Beijing Key Laboratory of Crop Genetic Improvement, China Agricultural University, Beijing 100193, China; gdd1990@cau.edu.cn (D.G.); wheatqilinghou@163.com (Q.H.); zhangrunqi91@sina.com (R.Z.); louhongyao123@163.com (H.L.); hebeiliyinghui@163.com (Y.L.); zhangyufeng@cau.edu.cn (Y.Z.); msyou67@cau.edu.cn (M.Y.); xiecj127@126.com (C.X.); liangrq@cau.edu.cn (R.L.); 2Beijing Engineering Research Center for Hybrid Wheat, Beijing Academy of Agricultural and Forestry Sciences, Beijing 100097, China; 3Institute of Evolution, University of Haifa, Mt. Carmel, Haifa 3498838, Israel

**Keywords:** *Triticum aestivum* L., *TaSPA*, transcriptome sequencing, prolamin, starch, wheat grains

## Abstract

Starch and prolamin composition and content are important indexes for determining the processing and nutritional quality of wheat (*Triticum aestivum* L.) grains. Several transcription factors (TFs) regulate gene expression during starch and protein biosynthesis in wheat. Storage protein activator (TaSPA), a member of the basic leucine zipper (bZIP) family, has been reported to activate glutenin genes and is correlated to starch synthesis related genes. In this study, we generated *TaSPA-B* overexpressing (OE) transgenic wheat lines. Compared with wild-type (WT) plants, the starch content was slightly reduced and starch granules exhibited a more polarized distribution in the *TaSPA-B* OE lines. Moreover, glutenin and ω- gliadin contents were significantly reduced, with lower expression levels of related genes (e.g., *By15*, *Dx2,* and ω-1,2 gliadin gene). RNA-seq analysis identified 2023 differentially expressed genes (DEGs). The low expression of some DEGs (e.g., *SUSase*, *ADPase*, *Pho1*, *Waxy*, *SBE*, *SSI*, and *SS II a*) might explain the reduction of starch contents. Some TFs involved in glutenin and starch synthesis might be regulated by *TaSPA-B,* for example, *TaPBF* was reduced in *TaSPA-B* OE-3 lines. In addition, dual-luciferase reporter assay indicated that both TaSPA-B and TaPBF could transactivate the promoter of ω-1,2 gliadin gene. These results suggest that *TaSPA-B* regulates a complex gene network and plays an important role in starch and protein biosynthesis in wheat.

## 1. Introduction

Bread wheat (*Triticum aestivum* L.), one of the three major food crops, provides a quarter of the world’s supply of plant proteins, carbohydrates, and dietary fiber [1,2]. In wheat grains, starch and prolamins are synthesized and stored in the endosperm. Starch synthesis involves a serious of complex and finely regulated enzymatic reactions and forms amylose and amylopectin [3]. The starch is stored as large A-type (average 10~35 µm diameter) and small B-type (<10 µm diameter) starch granules (SGs) [4]. Prolamins, composed of glutenin and gliadin, are synthesized on the rough endoplasmic reticulum (ER) and deposited to form protein bodies (PBs) [5]. PBs wrap around SGs and form two main abundant organelles in the endosperm of wheat grain. The quantities and proportions of starch and protein are key determinants of processing quality in wheat.

The special accumulation of prolamins in the endosperm during grain filling is controlled by several mechanisms. Regulation at the transcriptional level is the primary mode and depends on the interaction of with *cis*-motifs in the promoters of related genes, which is conserved in most cereals and dicots [6,7,8,9,10,11,12,13]. The bipartite endosperm box is composed of GCN4 motif, which is recognized by basic leucine zipper (bZIP) TFs, and P box, which is recognized by DNA binding with one finger (Dof) TFs [6,14,15]. B3 and MYB TFs also regulate prolamin genes [16,17,18,19]. Different TFs have been reported to establish a complex gene network via synergetic interactions and determine the expression of prolamin genes [17,20,21,22]. Post-transcriptional regulation is also important for prolamin accumulations. N-glycosylation, folding, and assembly of proteins often occur on the ER. Glutenin and gliadin are transported via Golgi-independent and dependent secretory pathways, respectively, and aggregate to form PBs in vacuoles [5]. Starch synthesis in cereals is catalyzed by a series of enzymes, including sucrose synthase (SUSase), invertase (INV), ADP-glucose pyrophosphorylase (ADPase), granule bound starch synthase (GBSS, Waxy), starch synthase (SS), starch branching enzyme (SBE), and starch de-branching enzyme (DBE) [3]. Up till now, only a few TFs that regulate starch synthesis have been identified. AP2, bZIP, and NAC TFs regulate starch synthesis related genes (SSRGs) in rice (*Oryza sativa*), maize (*Zea mays*) and wheat [11,12,23,24,25,26,27]. For example, maize Opaque-2 (O2) indirectly represses the TF gene *Prolamin-box binding factor* (*PBF*); furthermore, O2 and PBF interact to regulate gene networks for starch and protein biosynthesis [14,27,28]. Similar bZIP and Dof TF regulatory interaction is also found in rice [15,29,30].

Previous researches focused on the modification of the prolamin genes and starch synthesis genes to obtain ideal phenotypes [31,32]. With advances in genomics, the regulatory mechanism underlying the TFs and gene networks becomes a hot topic in wheat grain research [13,17,19,27,33,34]. Maize O2 is a typical TF with a broad influence, especially on starch and protein synthesis [28,35]. In wheat, the homolog *O2* gene *TaSPA,* was cloned as an activator of low molecular weight glutenin subunit (*LMW-GS*) gene [36]. Three homoeologous copies of *TaSPA* are located on the long arms of the homoeologous 1 group [37]. *TaSPA-B* activates high molecular weight glutenin subunit (*HMW-GS*) gene and was a candidate gene for a protein QTL [38,39,40]. In addition to *HMW-GS* and *LMW-GS*, the promoter of α- and γ- gliadin genes also contain the GCN4 motif [38,41,42,43]. However, up to date, *TaSPA-B* mediated regulatory role for starch and gliadins remains unknown.

In the current research, to investigate the role of *TaSPA-B* on the accumulation of starch and prolamin in wheat mature seeds, we generated *TaSPA-B* overexpressing (OE) transgenic wheat lines. In the *TaSPA-B* OE lines, starch and protein contents decreased and larger A-type SGs and smaller B-type SGs accumulated. RNA-seq analysis suggested that the differentially expressed genes (DEGs) regulated by *TaSPA-B* were involved in a wide range of metabolic pathways, especially starch and amino acid metabolism. Important TFs and key enzymes were identified as DEGs, which may contribute to the decrease of starch and protein contents. This study contributes a better understanding of the role of *TaSPA-B* in prolamin and starch accumulation in wheat grains.

## 2. Results

### 2.1. Generating TaSPA-B Overexpressing Lines

To examine the role of *TaSPA-B*, we transformed wheat with *TaSPA-B* under the *Glu-1Dx5* promoter, which could drive genes specifical expression in the wheat endosperm. We obtained six independent transformants, which were confirmed by PCR and sequencing analysis (Figure 1A). In the wild-type (WT) and *TaSPA-B* OE lines, the relative expression of *TaSPA* rapidly increased and reached a peak at 18 days post-anthesis (DPA), followed by decreasing gradually to 22 DPA (Figure 1B). At the peaking time, the expression of *TaSPA* in the three *TaSPA-B* OE lines was approximately 10- to 20-fold higher than in WT and remained high level at 22 DPA. These three lines were used to generate stable lines through PCR test, and their seeds were harvested and used in the following studies.

### 2.2. Morphology of Starch and Prolamin in Mature Seeds

We examined the morphology of starch, prolamin, and SGs in mature seeds by scanning electron microscopy (SEM). In wheat mature seeds, SGs were packed tightly with matrix proteins in the endosperm of mature seeds, varied in size, and were distributed randomly (Figure 2). Compared with WT, more and larger gaps and looser arrangement of PBs and SGs were observed in the three *TaSPA-B* OE lines (Figure 2A–D). The A-type SGs in WT appeared irregular round shape, while were more oval in the *TaSPA-B* OE lines, which might be caused by the looser endosperm structure (Figure 2E–H). The particle size of SGs showed a bimodal distribution with a peak at ~22 μm for A-type SGs and another peak at ~2 μm for B-type SGs in WT and *TaSPA-B* OE lines (Figure 3). There were more particles >22 μm and <2 μm in the *TaSPA-B* OE lines compared with WT, suggesting that larger A-type and smaller B-type SGs accumulated in these lines (Figure 3).

### 2.3. Starch and Prolamin Contents

Compared with WT, starch contents were slightly decreased in the three *TaSPA-B* OE lines (Figure 4A). HMW-GS (Bx14, By15, Dx2 and Dy12) and LMW-GS levels were dramatically reduced (Figure 4B). In addition, ω- gliadin levels were significantly reduced in the three *TaSPA-B* OE lines, α/β- gliadin levels were slightly reduced in OE-4 and OE-6, and γ- gliadin levels were significantly increased only in OE-4 lines (Figure 4C). On average, total glutenin content was significantly reduced by 40% in the three *TaSPA-B* OE lines, and total gliadin was only significantly reduced in OE-6 (Figure 4D). Previous result had shown that TaSPA-B activated glutenin genes, while our results suggested that overexpression of *TaSPA-B* reduced starch, glutenin, ω- and α/β- gliadin contents, even if the opposite was expected.

### 2.4. Relative Expression Analysis of TaSPA and Prolamin Genes

For further study whether the expression patterns of prolamin genes could be regulated by *TaSPA-B*, the immature seeds of WT and *TaSPA-B* OE-3 lines at 18 DPA (the expressing peaking time point of *TaSPA*) were collected for RNA extraction. The relative expressions of *TaSPA* and prolamin genes were measured by qRT-PCR. Expectedly, the *TaSPA-B* was significantly induced by 60-fold in *TaSPA-B* OE-3 lines. However, the *TaSPA-A* and *TaSPA-D* were slightly reduced (Figure 5). The expression of two glutenin genes, *Dx2* and *By15*, was significantly reduced, while the ω-1,2 gliadin gene was slightly reduced in the *TaSPA-B* OE-3 lines (Figure 5). The other measured prolamin genes *Bx14*, *Dy12*, *LMW*, α-, γ- and ω-5 gliadin showed no difference in expression.

### 2.5. Identification and Annotation of Differentially Expressed Genes in the TaSPA-B OE-3 Immature Seeds

For better understanding what pathways were regulated by *TaSPA-B* to decrease starch and protein content, we selected 18 DPA immature seeds of the *TaSPA-B* OE-3 line for RNA-seq analysis. We obtained 322,633,576 clean reads, with an average of 53,772,262 clean reads for WT and *TaSPA-B* OE-3 lines, with three repetitions (Appendix A). Q20 and Q30 values were >97.5% and 93.1%, respectively. On average, more than 70% of reads were uniquely aligned to the wheat reference genome of Chinese Spring (IWGSC RefSeq v1.1; Appendix A). The overlapped annotated genes were more than 90% in WT and *TaSPA-B* OE-3 lines, suggesting that the transcriptomes were similar in addition to expression level (Appendix A). A total of 2023 differentially expressed genes (DEGs) were identified, including 1437 up-regulated and 586 down-regulated genes in *TaSPA-B* OE-3 lines compared with WT (Appendix A).

In the *TaSPA-B* OE-3 lines, 873 up-regulated and 311 down-regulated genes were annotated to 565 Gene ontology (GO) terms (Appendix A). GO enrichment analysis indicated that some important terms could be affected by *TaSPA-B*, such as starch synthetic process, transferase activity, hydrolase activity and amyloplast (Figure 6A). Kyoto Encyclopedia of Genes and Genomes (KEGG) pathway analysis assigned 357 up-regulated and 175 down-regulated genes to 115 different pathways (Appendix A). KEGG enrichment analysis revealed 15 significantly enriched pathways, including some important pathways such as starch and sucrose metabolism, valine, leucine and isoleucine degradation, and phenylalanine metabolism (Figure 6B). 

### 2.6. DEGs Involved in Starch and Protein Metabolism

In the *TaSPA-B* OE-3 lines, the pyruvate orthophosphate dikinase (PPDK) and pyruvate dehydrogenase (PDH) complex genes were up-regulated, which suggested the enhanced nitrogen assimilation. Moreover, 27 key enzymes of starch synthesis were down-regulated (Table 1), which might explain the decrease in starch content (Figure 4A). The protein processing in ER relevant genes *Ribophorin I* (*RPN1*) and *B-cell receptor-associated protein* (*BAP31*) were down-regulated. *Ubiquitin-conjugating enzyme E2* and *ubiquitin ligase E3* were up-regulated (Table 1), which suggested the enhanced ubiquitin-mediated degradation. 

Among the DEGs, in addition to the over-expressed *TaSPA-B*, 130 TFs belonging to 22 TF families were identified in the *TaSPA-B* OE-3 lines (Appendix A). The TFs related to starch synthesis (e.g., *TaRSR1, TaGBF1* and *TabZIP*s) and protein (e.g., *TaSPA, TaPBF* and *TaSHP*) were differentially expressed (Table 2).

### 2.7. Dual-Luciferase Reporter Assay of the ω-1,2 Gliadin Gene Promoter

TaSPA-B and TaPBF-D have been identified to regulate wheat glutenin gene, and their homologs in maize (O2 and PBF) show stimulatory and additive effects on maize zein and wheat glutenin genes [14,34,38]. In our results, ω-1,2 gliadin gene and *TaPBF* showed a little reduction in *TaSPA-B* OE-3 lines compared with WT (Figure 5, Table 2). Therefore, we performed the dual-luciferase reporter (DLR) assay to determine whether TaSPA-B and TaPBF-D could regulate ω-1,2 gliadin gene. In this system, 35S-SPA-B and 35S-PBF-D were constructed as effectors (Figure 7A). The ω-1,2 gliadin gene promoter was fused with LUC to generate the reporter ω-1,2-LUC. Compared with the negative control, 35S-SPA-B and 35S-PBF-D increased the LUC activity by 2- and 11-fold, respectively (Figure 7B). Moreover, the combination of two effectors (SPA-B&PBF-D) resulted in 20-fold increase in LUC activity compared with negative control (Figure 7B). The results indicated that both TaSPA-B and TaPBF-D could activate the ω-1,2 gliadin gene promoter. TaPBF-D showed a higher activation than TaSPA-B and they provide additive activation of ω-1,2 gliadin gene. 

## 3. Discussion

The bZIP family TFs play roles in various biological processes and stress responses. The O2 subfamily, including O2 and RISBZ1, is a typical bZIP family and family members interact with the GCN4 motif to regulate prolamin genes and SSRGs in maize and rice [23,30,44,45]. The *O2* mutant affects starch and zein accumulation [28,35]. *TaSPA-B*, which is a homolog of *O2*, was isolated as an activator of *LMW-GS* and *HMW-GS* [36,38]. An association study reported that the three homoeologous copies of *TaSPA* were strongly associated with prolamin composition [46]. Interestingly, correlation analysis of gene expression revealed that *TaSPA-B* was also correlated with SSRGs [47]. To understand the effect of *TaSPA-B* on the accumulation of prolamin and starch, we developed three *TaSPA-B* OE lines. The expression pattern of *TaSPA* showed the trend of first increasing and then decreasing from 10 to 22 DPA, which was consistent with the previous finding [37]. The *TaSPA* in the *TaSPA-B* OE lines showed a similar expression pattern as in WT, but with higher transcriptional level at four time points (Figure 1B).

In wheat grains, SGs and PBs distribute randomly in the starchy endosperm. The A-type SGs form earlier and larger than B-type SGs [48], but account for <10% of the numbers of SGs [49]. PBs accumulate different compositions of glutenin and gliadin in different endosperm tissues and arrange compactly with SGs [50]. In the *TaSPA-B* OE lines, SGs and protein matrix arranged looser in mature seeds, compared with WT (Figure 2). The SGs in the *Triticeae* grass family show unique bimodal granule morphology [48], and this was also observed in our results (Figure 3). However, the SGs showed a more polarized distribution and starch content showed a slight decrease in the *TaSPA-B* OE lines compared with WT (Figure 3 and Figure 4A). It has been proved that TaSPA-B activates *LMW-GS* and *HMW-GS* [36,38,39], and is associated with glutenin content [46]. On the contrary, our results showed that HMW-GS (Bx14, By15, Dx2 and Dy12) and LMW-GS contents were reduced dramatically in the *TaSPA-B* OE lines (Figure 4B). Moreover, the ω-1,2 and α/β- gliadin contents also showed a slight decrease in the *TaSPA-B* OE lines. The relative expression analysis revealed that the levels of *By15, Dx2* and ω-1,2 gliadin gene were reduced, which might explain the decrease of protein contents (Figure 4 and Figure 5). Interestingly, other prolamin genes had no significantly transcriptional change in spite of the reduction of accumulations. We speculated that *TaSPA-B* might affect some important genes at the transcriptional and post-transcriptional level to play an important role in starch and protein synthesis.

Our RNA-seq analysis identified a wide range of metabolic pathways regulated by *TaSPA-B* (Figure 6), especially starch and amino acid metabolism, which might explain the content changes of starch and protein in *TaSPA-B* OE-3 lines (Figure 4B,C). PPDK play an important role in the balance of starch and protein as the central substance by regulating the ratio of phosphoenolpyruvate and pyrophosphoric acid in maize and rice [51,52,53]. PDH catalyzes a rate-limiting step of pyruvate entering the tricarboxylic acid cycle (TCA). The up-regulation of *PPDK* and *PDH* genes might enhance nitrogen assimilation in *TaSPA-B* OE-3 lines, however, 27 key enzymes of starch synthesis were down-regulated, which might cause the decrease in starch content (Table 1, Figure 4A). Moreover, the *plastidial phosphorylase* (*Pho1)* mutation increased the accumulation of small SGs and modified the amylopectin structure in rice [54]. We also found an increase in small B-type SGs and the down-regulation of *Pho1* in *TaSPA-B* OE-3 lines, perhaps *Pho1* have a similar effect (Figure 3, Table 1). Some important genes related to protein processing in ER were also down-regulated, while the ubiquitin-proteasome complex related genes were up-regulated.

Maize O2 regulates a complex gene network, such as genes coding starch synthesis related enzymes, cyPPDK1, cyPPDK2 and PBF, to balance starch and protein synthesis [27,28,35]. The expression of *TaRSR1* and *TabZIPs* is correlated to the expression of SSRGs in wheat [9,13,25,47]. However, the overexpressed *TaSPA-B* was negatively correlated, *TaRSR1* was positively correlated to 27 SSRGs in the *TaSPA-B* OE-3 lines, showing not completely in line with the previous results [9,25,47]. Other TF genes related to SSRGs, such as the negative regulatory factor genes *TabZIP151, TabZIP194.3* and *TabZIP229.1* (up-regulated), and the positive regulatory factor gene *TabZIP167.2* (down-regulated), were also differentially expressed (Table 2) [13]. A series of TFs, including TaSPA, TaGAMYB, TaFUSCA3, TaSHP, and TaPBF-D, regulate the glutenin genes in wheat [17,19,33,34,36,38]. In the *TaSPA-B* OE-3 lines, *TaSPA-A* and *TaSPA-D* were down-regulated (Figure 5); *TaSHP*, encoding a transcriptional repressor of glutenin genes, was up-regulated; *TaPBF*, encoding a transcriptional activator of glutenin genes was down-regulated (Table 2). The expression changes of these TFs genes corresponded to the changes in starch and protein contents.

Maize *PBF* is an indirect O2 target gene and is up-regulated in the B73*O2* mutant, suggesting the negative role of O2 to *PBF* [28]. Importantly, PBF and O2 can interact to affect the accumulation of zein and starch [14,28]. Consistent with these results, *TaPBF* was reduced in *TaSPA-B* OE-3 lines (Table 2). TaPBF activated glutenin genes by regulating methylation of the promoter [34]. Our DLR assay revealed that TaSPA-B and TaPBF-D could activate the expression of ω-1,2 gliadin gene, and TaPBF-D showed a higher activation than TaSPA-B (Figure 7B). The down-regulation of *TaPBF* might be one of factors for prolamin reduction. Therefore, *TaSPA-B* regulates a complex gene network and plays an important role in starch and protein biosynthesis in wheat.

## 4. Materials and Methods

### 4.1. Plant Materials

Wheat cultivar Fielder and Jing 411 were planted in the greenhouse at 26/20 °C (day/night) with 75% relative humidity and a light intensity of 40 µmol m^−2^ s^−1^. Mature seeds were harvested and ground into flour with an experimental mill (Brabender, Duisburg, Germany). *Nicotiana benthamiana* seeds were planted in the greenhouse at 22/18 °C (day/ night) with 60% relative humidity and a light intensity of 3000 lux. Young leaves were used for DLR assay.

### 4.2. DNA Extraction, RNA Extraction, cDNA Synthesis, and qRT-PCR

Genomic DNA was isolated from one-week-old wheat leaves using the CTAB method. Seeds were harvested at 10, 14, 18 and 22 DPA, snap-frozen in liquid nitrogen, and stored at −80 °C. Total RNA was extracted from seed samples using the RNA prep Pure Plant Kit (Tiangen Biotech, Beijing, China) and reverse transcribed to cDNA using the PrimeScript^TM^ RT reagent Kit with gDNA Eraser (Takara, Dalian, China). The qRT-PCR primers were listed in Appendix A. *GAPDH* was the house-keeping gene used as internal control. The primer of *TaSPA* was designed to amplify three homoeologous copies according to the conserved sequence. The primers of *LMW-GS*, *α-*, *γ-*, *ω-1,2* and *ω-5 gli* were designed to amplify the whole multigene family members according to the conserved sequences. The qRT-PCR reaction system was 20 μL according to the manufacturer’s instructions of SYBR Premix Ex TaqII (Takara, Dalian, China). The recommended PCR amplification procedure for Applied Biosystems 7500 Real-Time PCR System (Applied Biosystem, Foster City, CA) was used for analysis. The relative expression levels of genes were calculated using the 2^−ΔΔCt^ method and normalized to *GAPDH*. Three independent experiments were performed with three technical repetitions for each sample.

### 4.3. Plasmid Construction and Genetic Transformation for Wheat

The full-length coding sequence of *TaSPA-B* was amplified from Jing411 cDNA and inserted in the pCAMBIA3300 vector via restriction enzyme digestion and ligation. The primer for *TaSPA-B* was listed in Appendix A. The *Bar* gene of the pCAMBIA3300 vector was driven by CaMV35S. The *Glu-1Dx5* promoter was digested from the vector pBAC47p and ligated to the vector before *TaSPA-B* to ensure efficient expression in wheat endosperm [55]. The vector was transferred into Agrobacterium strain C58C1. Immature Fielder wheat embryos were used for agrobacterium-mediated transformation according to the modified protocol developed by Japan Tobacco Company [56]. The stable integration of *TaSPA-B* was confirmed by PCR test of genomic DNA. Two pairs of primers were designed for detecting the *TaSPA-B* and *Bar* in the vector (Appendix A).

### 4.4. Extraction and Size Distribution of Starch Granules, and Morphology Observation

The SGs were isolated according to Peng et al. [4]. The starch pellets were dried at room temperature and stored at −20 °C. The particle size distribution of SGs was measured with a Mastersize 2000 Laser Diffraction instrument (Malvern Instruments Ltd., Worcestershire, UK). Dried seeds were sliced down the middle to create transverse sections and adhered to an aluminum stub with conductive double-sided tape. The isolated SGs were gently transferred onto the stub. The samples were sprayed with a thin film of gold (10 nm) and observed under a scanning electron microscope (Hitachi S3400N, Tokyo, Japan). For WT and *TaSPA-B* OE lines, three independent mature seeds with three transverse sections were imaged.

### 4.5. Starch and Protein Extraction and Content Analysis

The total starch content of flour was measured using a Total Starch Assay kit (Megazyme, Wicklow, Ireland) according to the manufacturer’s protocol. The sequential extraction of gliadin and glutenin from wheat flour and reversed-phase ultra-performance liquid chromatography (RP-UPLC) were performed according to Han et al. [57] and Yan et al. [58,59] with slight modifications. Wheat flour (0.1 g) was mixed with 1 mL 50% n-propanol and incubated at 65 °C for 30 min. Three times repeated supernatants were combined and subjected to gliadin detection. 250 μL solution A (50% n-propanol, 20% 1 M pH 6.8 Tris-HCl and freshly added 1% dithiothreitol) and 250 μL solution B (50% n-propanol, 20% 1 M pH 6.8 Tris-HCl and freshly added 1.4% 4-vinyl pyridine) were successively added to the residues, followed by incubation at 65 °C for 30 min after each addition. Glutenin was precipitated with 60% acetone, washed with ethanol, and dissolved in 100 μL solution (50% acetonitrile and 0.5% trifluoroacetic acid). Both protein solutions were filtered through 0.45 μm nylon filters. RP-UPLC was performed on an Acquity UPLC (Waters Crop., Milford, MA, USA) with an Agilent ZORBAX 300SB-C18 column (4.6 × 150 mm, 5 μm, Agilent Technologies, Palo Alto, CA, USA). The parameters were absorbance at 210 nm, column temperature of 60 °C, flow rate of 1 mL/min, and 4 μL injection volumes. The elution buffer was consisted of acetonitrile and 0.06% trifluoroacetic acid. The gradient was gradually increased from 25 to 50% in 25 min for gliadin and from 21 to 53.5% in 50 min for glutenin. Each sample was sequentially injected three times for technical replicates. 

The integration of peak areas was performed automatically with the same criteria to obtain effective peaks and avoid noise peaks. The effective peak areas were used to represent protein contents. The ω-, α/β-, and γ- gliadins were eluted from 6 to 13 min, from 13 to 20 min, and from 20 to 30 min, respectively. The four subunits of HMW-GS were eluted at approximately 15, 18, 20 and 22 min for Dy12, By15, Dx2, and Bx14, respectively, and LMW-GSs were eluted from 26 to 36 min. Every sample was compared with WT, respectively, with three repetitions.

### 4.6. RNA-Seq Analysis

The RNA-seq was performed using the 18 DPA immature seeds of Fielder and *TaSPA-B* OE-3 lines. Each sample included three biological repetitions. The total RNA was extracted and detected by agarose gel electrophoresis and Agilent 2100 Bioanalyzer. The concentration and purity of the RNA were checked with a NanoDrop 2000. 1 μg RNA was used for library construction with the Truseq RNA sample prep Kit (Illumina, San Diego, CA, USA). The pair-end sequencing was performed on the Illumina Novaseq 6000 platform. The adapter sequences and low-quality sequences of raw reads were removed with the SeqPrep (https://github.com/jstjohn/SeqPrep) and Sickle (https://github.com/najoshi/sickle) software to obtain clean reads. The TopHat2 (Version 2.1.1) was used to map the clean reads to the Chinese Spring (CS) wheat reference genome (http://plants.ensembl.org/Triticum_aestivum/Info/Index). The reads with one single aligned position were used for further analysis. The Transcript per Million (TPM) values were used as the variables to measure the expression levels of transcripts and to eliminate the impact of differences in gene length and sequencing volume on the calculated gene expression [60,61]. The gene functions were annotated to the NR (NCBI non-redundant protein sequences), Swiss-Prot (a manually annotated and reviewed protein sequence database), Pfam (Protein family), EggNOG (Evolutionary Genealogy of Genes: Non-supervised Orthologous Groups), GO and KEGG databases. DEGs were identified using the criteria |log_2_FC|>1 and FDR < 0.05 with DESeq2 Software (version 1.24.0). GO and KEGG enrichment of DEGs was implemented with Goatools (version 0.6.5) with BH method. Transcription factor prediction for DEGs was performed with PlantTF 4.0 (http://planttfdb.cbi.pku.edu.cn/).

### 4.7. Dual-Luciferase Reporter Assay of Gliadin Gene Promoter Activations

DLR assay was performed as described by Liu et al. [62]. To generate the 35S-SPA-B and 35S-PBF-D effector constructs (driven by the CaMV 35S promoter), we ligated the coding sequence of *TaSPA-B* and *TaPBF-D* to the PHB vector, respectively. The ~1 kb promoter sequence of the ω-1,2 gliadin gene was amplified from Fielder genomic DNA and used to drive the *LUC* of pGreenII 0800-LUC. The *Ren* driven by the CaMV 35S promoter was used as an internal control. The primers used for the construction of vectors were listed in Appendix A. The constructed vectors were transferred into Agrobacterium strain GV3101 (pSoup) and injected into young leaves of four- to six-week-old *Nicotiana benthamiana* plants. Forty-eight hours later, the ratio of LUC/Ren activities was measured according to the Dual-luciferase assay reagents (Promega, CA, USA) with a Synergy 2 Multi-Detection Microplate Reader (BioTek Instrument Inc., Winooski, VT, USA) [63].

### 4.8. Statistical Analysis

Statistical analysis of significant differences was performed using Student’s *t*-test. The sequence alignment was performed using the DNAMAN software (version 8.0). The bar and line charts were graphically drawn using Origin 9.1 (OriginLab) software.

## Figures and Tables

**Figure 1 ijms-21-03257-f001:**
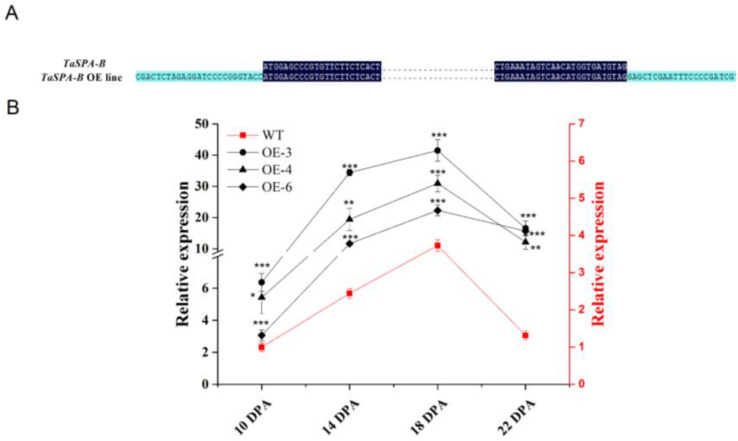
Validation of the *TaSPA-B* overexpressing (OE) lines in the T_1_ generation. (**A**) Partial sequence alignment of the vector and *TaSPA-B* in the *TaSPA-B* OE lines. The dark blue shows the sequence of *TaSPA-B*. The light blue shows the sequence of the vector. The points represent the omitted base sequences. (**B**) Relative expression level of *TaSPA* at 10, 14, 18, and 22 days post-anthesis (DPA) endosperms in the T_1_ generation measured by quantitative reverse-transcription PCR (qRT-PCR) analysis. The left black y-axis shows the *TaSPA* expression level in the *TaSPA-B* OE lines, and right red y-axis shows the *TaSPA* expression level of wild-type (WT). The relative expression represents the three homoeologous copies of *TaSPA.* The values are the mean of three biological repetitions and error bars represent the SD. The gene expression in the *TaSPA-B* OE lines at each period is compared with WT, respectively. Asterisks indicate significant differences (Student’s *t*-test, * *p* < 0.05; ** *p* < 0.01; *** *p* < 0.001).

**Figure 2 ijms-21-03257-f002:**
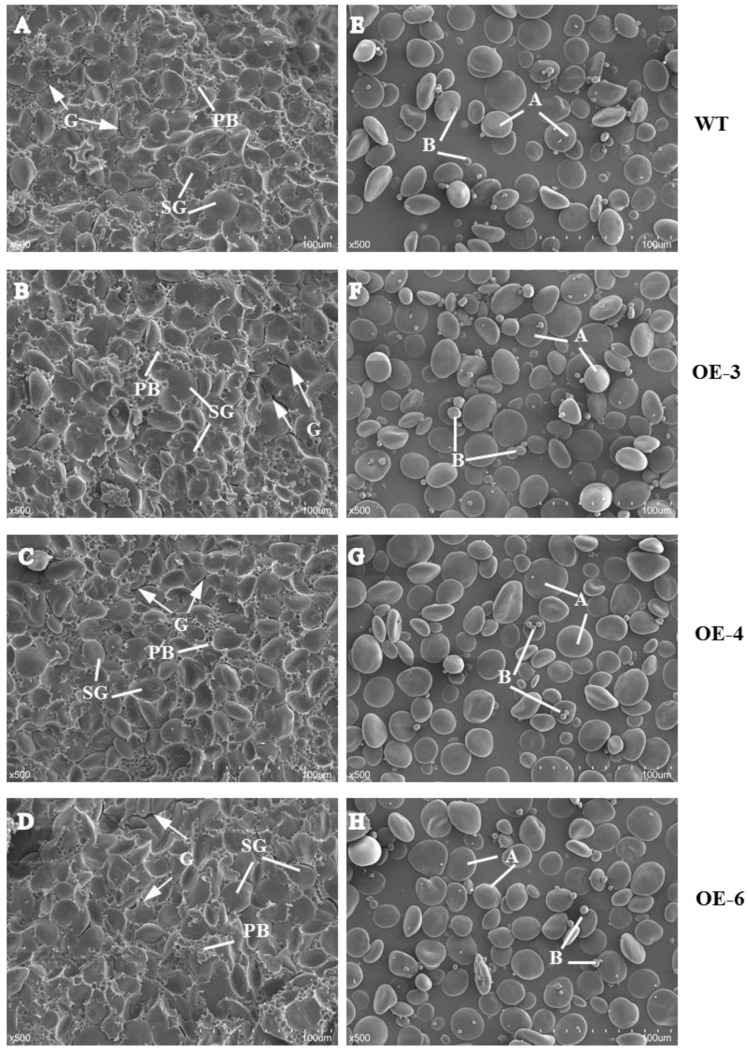
Scanning electron micrograph of mature seeds and isolated starch granules in WT and *TaSPA-B* OE lines. (**A**–**D**) Transverse sections of mature seeds. (**E**–**H**) Isolated starch granules. PB, protein body; SG, starch granule; A, A-starch granule; B, B-starch granule; G, the gap between PB and SG. Arrows indicate the gaps between starch granules and protein bodies. Scale bars, 100 μm (500× magnification).

**Figure 3 ijms-21-03257-f003:**
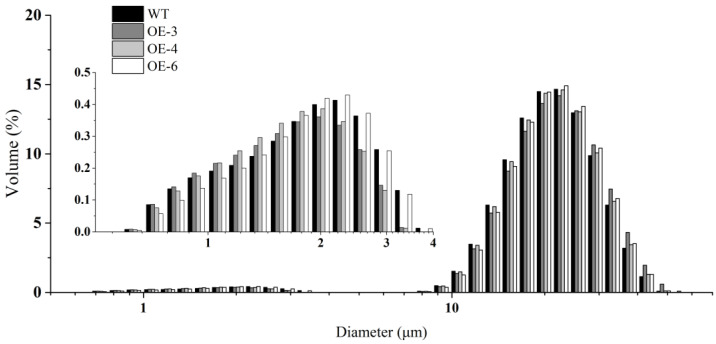
The particle size distribution of SGs. The inset shows the distribution of B-type SGs.

**Figure 4 ijms-21-03257-f004:**
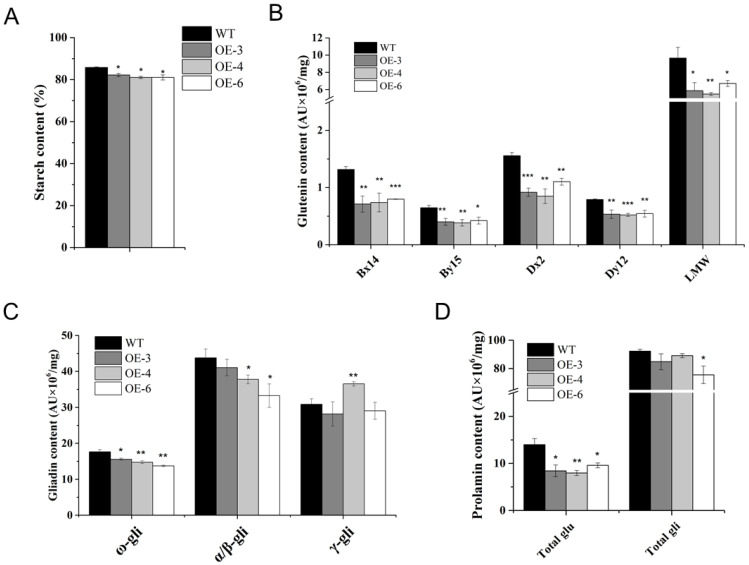
Starch and prolamin contents in WT and *TaSPA-B* OE lines. (**A**) Starch contents. (**B**) Glutenin contents. Bx14, By15, Dx2 and Dy12 are the four subunits of HMW-GS. (**C**) Gliadin contents. ω-, α/β- and γ-gli represent the compositions of gliadin. (**D**) Total glutenin and gliadin contents. The values are the mean of three biological repetitions and error bars represent the SD. Asterisks indicate significant differences compared with WT (Student’s *t*-test, * *p* < 0.05; ** *p* < 0.01; *** *p* < 0.001).

**Figure 5 ijms-21-03257-f005:**
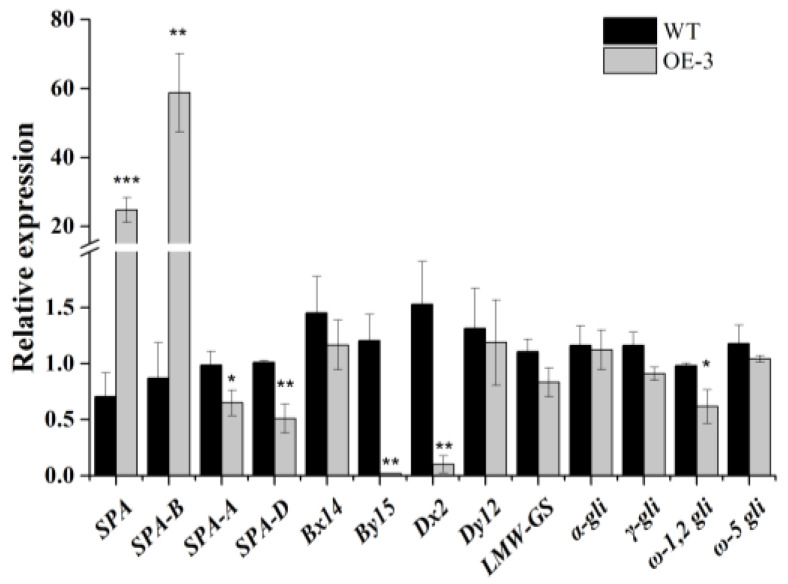
The relative expressions of *TaSPA* and prolamin genes in 18 DPA endosperm of WT and *TaSPA-B* OE-3 lines. *SPA* represents the three homoeologous copies and *SPA-A, -B* and *-D* represent each single homoeologous copy of *TaSPA*. *Bx14*, *By15*, *Dx2* and *Dy12* represent the coding genes of the four subunits of HMW-GS. *α*-, *γ*, *ω-1,2* and *ω-5* gli represent the coding genes of different components of gliadin. The values are the mean of three biological repetitions and error bars represent the SD. Asterisks indicate significant differences (Student’s *t*-test, * *p* < 0.05; ** *p* < 0.01; *** *p* < 0.001).

**Figure 6 ijms-21-03257-f006:**
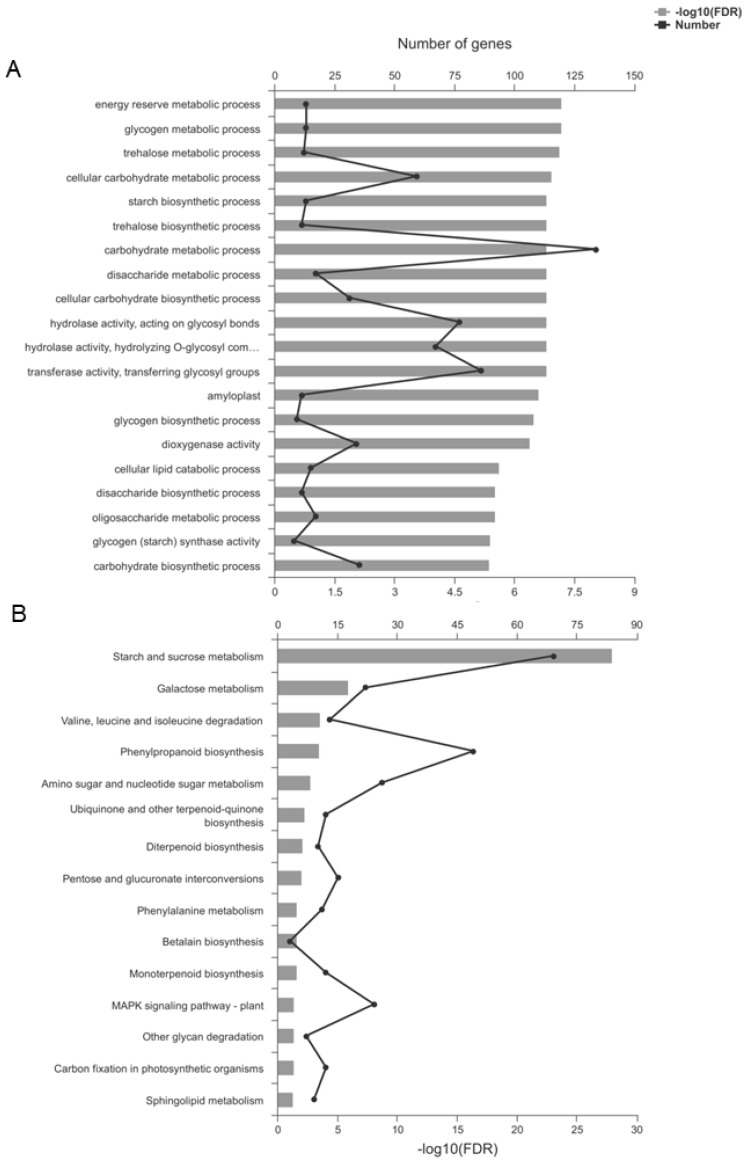
Gene ontology (GO) and Kyoto Encyclopedia of Genes and Genomes (KEGG) enrichment of differentially expressed genes (DEGs). (**A**) The top 20 enriched GO terms. (**B**) The 15 enriched KEGG pathways. FDR represents the adjusted P-value using the Benjamini–Hochberg (BH) method.

**Figure 7 ijms-21-03257-f007:**
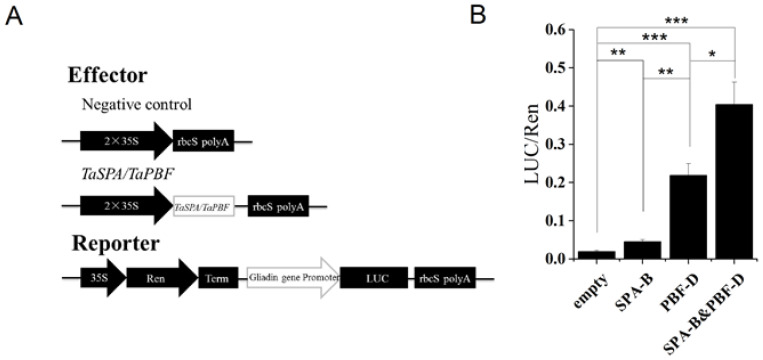
The activation of ω-1,2 gliadin gene promoter by TaSPA-B and TaPBF in *N. benthamiana* leaves. (**A**) Schematic diagram of effectors and reporter. Ren, Renilla luciferase; LUC, firefly luciferase; Term, terminator. (**B**) The relative LUC activities driven by the promoter of ω-1,2 gliadin gene. The ratios of LUC/Ren represent the relative promoter activities. Empty, negative control effector; SPA-B, 35S-SPA-B effector; PBF-D, 35S-PBF-D; SPA-B&PBF-D, combination of 35S-SPA-B and 35S-PBF-D. The values are the mean of three biological repetitions and error bars represent the SD. The LUC activities of SPA-B, PBF-D and the combination of effectors (SPA-B&PBF-D) are compared with the empty, respectively. The LUC activity of PBF-D is compared with SPA-B and the combination of effectors (SPA-B&PBF-D), respectively. Asterisks indicate significant differences (Student’s *t*-test, * *p* < 0.05; ** *p* < 0.01; *** *p* < 0.001).

**Table 1 ijms-21-03257-t001:** Representative DEGs involved in starch and protein synthesis.

Gene ID	Log_2_FC	Gene Name	Annotation
TraesCS1B02G264900	1.48	*PPDK*	pyruvate orthophosphate dikinase
TraesCS1D02G252900	1.22	*PPDK*	pyruvate orthophosphate dikinase
TraesCS1A02G099500	1.13	*pdhD*	pyruvate dehydrogenase complex
TraesCS5A02G476700	2.40	*pdhC*	pyruvate dehydrogenase complex
TraesCS5B02G116300	1.31	*pdhC*	pyruvate dehydrogenase complex
TraesCS5D02G126000	1.55	*pdhC*	pyruvate dehydrogenase complex
TraesCS4A02G446700	−1.42	*SUSase*	sucrose synthase
TraesCS2D02G403600	−3.84	*SUSase*	sucrose synthase
TraesCS2D02G175600	−3.56	*SUSase*	sucrose synthase
TraesCS2A02G406700	−2.27	*SUSase*	sucrose synthase
TraesCS2A02G168200	−2.62	*SUSase*	sucrose synthase
TraesCS2B02G194200	−3.02	*SUSase*	sucrose synthase
TraesCS5D02G182600	−1.42	*ADPase*	ADP-glucose pyrophosphorylase
TraesCS5D02G484500	−1.72	*ADPase*	ADP-glucose pyrophosphorylase
TraesCS5A02G472000	−1.68	*ADPase*	ADP-glucose pyrophosphorylase
TraesCS1B02G449700	−1.05	*ADPase*	ADP-glucose pyrophosphorylase
TraesCS7A02G287400	−1.19	*ADPase*	ADP-glucose pyrophosphorylase
TraesCS7D02G064300	−1.44	*WAXY,GBSSI*	granule bound starch synthase
TraesCS7A02G549300	−1.64	*GBE1,SBE*	starch branching enzyme
TraesCS2A02G310300	−1.33	*GBE1,SBE*	starch branching enzyme
TraesCS7A02G549100	−1.24	*GBE1,SBE*	starch branching enzyme
TraesCS2D02G308600	−1.63	*GBE1,SBE*	starch branching enzyme
TraesCS7D02G535600	−2.68	*GBE1,SBE*	starch branching enzyme
TraesCS7B02G472300	−2.41	*GBE1,SBE*	starch branching enzyme
TraesCS7B02G472500	−1.09	*GBE1,SBE*	starch branching enzyme
TraesCS7D02G117800	−1.22	*SSI*	soluble starch synthase
TraesCS7A02G120300	−1.12	*SSI*	soluble starch synthase
TraesCS7B02G093800	−1.54	*SSII a*	soluble starch synthase
TraesCS7A02G189000	−1.43	*SS II a*	soluble starch synthase
TraesCS7D02G190100	−1.37	*SSII a*	soluble starch synthase
TraesCS5A02G395200	−1.98	*Pho1*	Alpha-1,4 glucan phosphorylase
TraesCS5B02G400000	−1.48	*Pho1*	Alpha-1,4 glucan phosphorylase
TraesCS5D02G404500	−1.70	*Pho1*	Alpha-1,4 glucan phosphorylase
TraesCS5B02G550300	−1.31	*BAP31*	B-cell receptor-associated protein
TraesCS4A02G334800	−1.11	*BAP31*	B-cell receptor-associated protein
TraesCS2B02G616300	−2.77	*OST1, RPN1*	Ribophorin I
TraesCS2D02G566500	−8.95	*OST1, RPN1*	Ribophorin I
TraesCS4A02G379700	−1.45	*SKP1*	SKP1-like protein 1
TraesCS1A02G133100	−1.10	*HSPA1s*	heat shock 70 kDa protein
TraesCS7B02G083100	−1.81	*HSP20*	heat stress protein 20
TraesCS5A02G257700	−1.21	*HSP20*	heat stress protein 20
TraesCS5D02G266000	−1.31	*HSP20*	heat stress protein 20
TraesCS7D02G179000	3.94	*HSP20*	heat stress protein 20
TraesCS5A02G511800	6.42	*BIP*	binding protein
TraesCS3A02G537600	1.37	*DNAJC3*	DnaJ homolog subfamily C member 3
TraesCS3D02G543100	1.08	*DNAJC3*	DnaJ homolog subfamily C member 3
TraesCS3D02G164900	1.38	*UBE2O*	ubiquitin-conjugating enzyme E2 complex
TraesCS1A02G094100	3.31	*SIAH1*	ubiquitin-protein ligase E3 complex
TraesCS1D02G102700	2.32	*SIAH1*	ubiquitin-protein ligase E3 complex
TraesCS3A02G288900	1.47	*RCHY1*	ubiquitin-protein ligase E3 complex
TraesCS3A02G527600	1.50	*CDH1*	ubiquitin-protein ligase E3 complex
TraesCS5A02G177100	2.26	*UBE2D*	ubiquitin-conjugating enzyme E2 complex

Note: FC represents the fold change of expression in *TaSPA-B* OE-3 lines compared with WT. “−” represents own-regulation.

**Table 2 ijms-21-03257-t002:** Representative differentially expressed transcription factors (TFs) involved in starch and protein synthesis.

Gene ID	Log_2_FC	Gene Name
TraesCS1B02G343500	5.38	*TaSPA*
TraesCS1D02G332200	−1.51	*TaSPA*
TraesCS5A02G155900	−2.05	*TaPBF*
TraesCS5B02G154100	−2.19	*TaPBF*
TraesCS5D02G161000	−2.16	*TaPBF*
TraesCS5A02G440400	1.07	*TaSHP*
TraesCS5D02G447500	1.21	*TaSHP*
TraesCS5B02G444100	1.04	*TaSHP, TabZIP151* ^1^
TraesCS7B02G114300	3.69	*TabZIP229. 1* ^1^
TraesCS5D02G178800	−1.02	*TabZIP167.2* ^1^
TraesCS1B02G076300	−2.18	*TaRSR1*
TraesCS1A02G058400	−2.63	*TaRSR1*
TraesCS1A02G409800	1.08	*TaGBF1*
TraesCS1B02G439800	1.20	*TaGBF1*
TraesCS1D02G417100	1.15	*TaGBF1*
TraesCS7A02G488600	5.84	*TabZIP194.3* ^1^
TraesCS7B02G391800	4.18	*TabZIP206* ^1^
TraesCS7D02G475100	6.11	*TabZIP217.1* ^1^

Note: ^1^ The bZIP identifiers are described by Kumar et al. (2018) [13]. FC represents the fold change of expression in *TaSPA-B* OE-3 lines. “−” represents down-regulation.

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
