# Peer review of "Over-Expressing TaSPA-B Reduces Prolamin and Starch Accumulation in Wheat (Triticum aestivum L.) Grains"

_ijms, 2020, doi:10.3390/ijms21093257_

Round 1
Reviewer 1 Report
In this manuscript, it is reported the over-expression of a transcription factor implicated in the activation of glutenin genes, and it is also correlated with starch synthesis. Wheat grain protein fractions were significantly altered as well as starch content. RNA-seq analysis was carried out to find out differentially expressed genes.
Results reported are of interest for those working in the field. However, several shortcomings and mistakes have been found in the manuscript
The objective of the work is not clear in the manuscript, and I think it would be important to give a working hypothesis, an objective. As it stands now, it seems that the aim was to over-express the TF and see what happens.
In the introduction section
At the end of the second paragraph: I think you want to write “TFs” instead of “IFs” in “AP2, bZIP, and NAC IFs regulate starch synthesis related genes (SSRGs) in rice and maize …”.
In the result section:
In the first paragraph: it is repeated “expression” in “the expression of TaSPA expression in…”.
In section 2.2: Citing Figure 2E-F, did you want to cite Figure 2E-H instead?
In section 2.5: Citing Figure 6B in the GO results part, did you want to cite Figure 6A instead?
In the caption of Figure 7: the meaning of ** is not described. It is also confused what the comparisons are in the Student’s t-test analysis, are you comparing all with the wt ?, this must be clarified for this Figure and others.
In Figure 1B, for wild-types values, there are no error bars. Could you clarify if the standard deviation is too small to see it in the figure?
In the caption of Figure 3, you wrote that the insert shows the distribution of A-type SGs, but on the x-axis, the maximum diameter value is 4 µm, did you want to write B-type instead?
Interestingly, the glutenin and ω-1,2 and α/β-gliadins decrease in OE lines, even if the opposite was expected with TaSPA-B overexpressed. could you address this result?
In the discussion section:
In the third paragraph: the acronym TCA is not defined.
In the fourth paragraph: the verb of the phrase is missing: “Other TF genes related to SSRGs were also differentially expressed, …”.
In the fourth paragraph: “While” in the “While the overexpressed TaSPA-B were negatively …” does not make sense in the way this phrase is presented. Maybe you wanted to say: “While the overexpressed TaSPA-B was negatively correlated, TaRSR1 was positively correlated to 27 SSRGs, showing not completely in line with the previous results …”.
There is no mention of the down-regulation of the homeologs of TaSPA-B: TaSPA-A and TaSPA-D at 18 DPA in OE-3 line.
It might be interesting to discuss whether the prolamin genes in RNA-seq data analysis behave the same as in qRT-PCR, and if they are differentially expressed in DEG analysis for OE line.
In the material and methods section:
The acronyms of DPA, GO and KEGG, as far as I have seen, are defined for the first time in this section but appear in the previous ones.
What did you use as standard in RP-UPLC analysis?
The FPKM normalization method used for transcript expression is not appropriate for inter-sample comparisons in DEG analysis. It is recommended to use Transcript per Million (TPM), Trimmed Mean of M-values (edgeR) or the median of ratio normalization (DESeq2) methods, among others. The first one is usually used in reference papers as Borrill et al. (2019, Borrill, P., Harrington, S. A., Simmonds, J., & Uauy, C. (2019). Identification of transcription factors regulating senescence in wheat through gene regulatory network modelling. Plant physiology, 180(3), 1740-1755.), Ramírez-González et al. (2018, Ramírez-González, R. H., Borrill, P., Lang, D., Harrington, S. A., Brinton, J., Venturini, L., ... & Khedikar, Y. (2018). The transcriptional landscape of polyploid wheat. Science, 361(6403), eaar6089.). The reason is that in TPM normalization gene length is normalized first, and sequencing depth is normalized second. So, the sum of TPM values of all genes in each sample are the same.
In the 4.7 section: the verb is missing in “The primers used for the construction of vectors listed in Table S8.”, “are listed”.
I do not understand the gene function annotation process. Did you use the IWGSC RefSeq Annotation v1.1?. Why are there differences in the number of genes annotated for WT and OE lines in Figure S1A? In the results section, where this Figure is cited, it says that more than 92% annotated genes were co-expressed (I assumed this is the intersection in Venn diagram), what did you mean by co-expressed, not DEG genes?
In the reference section:
I have found a repeated reference: 6 and 17
Reviewer 2 Report
The authors of this manuscript generated some transgenic wheat lines overexpressing (OE) TaSPA ‐ B, a transcription factor regulating starch and protein content and which may potentially affect grain quality. The study is quite complete, because it investigated several mutants features including: chemical composition, starch and protein bodies morphology, relative expression of TaSPA and prolamin genes. Through RNA-seq analysis the authors identified a total of 2023 differentially expressed genes and attempted to identify which pathways were regulated by TaSPA-B. Finally dual luciferase reports essay of the gene promoter demonstrated that the down-regulated TaPBF might generate the lower content and gene expression of omega-1,2 gliadin.
Identification and characterization of transcription factors affecting protein and starch accumulation in the cereal grain is very important, but is very complex, because it involves temporally intertwined biochemical processes. For this reason, microscopic and biochemical analyses and gene expression tests should be observed at the same physiological stage. It seems that gene expression was made at 18DPA, the microscopic examination was made on mature seeds and the stage of starch and prolamin analysis does not seem to me to be indicated in the text.
The authors should better describe how many transverse section of mature seeds were made. Has a computerized analysis of the granules number and areas been made?
The description of the regulatory sequences of the Bar genes has perhaps been described in the supplementary material, but was not in my draft
The great number of differentially expressed genes identified by the authors of this work proves the difficulties in establishing which pathways, regulated by TaSPA-B, above all affected the starch and protein biosynthesis. Hence the authors should avoid to assess that that the lower content and gene expression of omega-1,2 gliadin can be ascribed to the down-regulated TaPBF.
